# Modified nnU-Net for the MICCAI KiTS21 Challenge

Xu Lizhan[1], Shi Jiacheng[1], and Dong Zhangfu[1]

[1]Southeast University

213172291@seu.edu.cn

**Abstract.**

KiTS21 Challenge is to develop the best system for automatic semantic segmentation of renal tumors and surrounding anatomy. The organizers provide a dataset of 300 cases and each case's CT scan is segmented to three semantic classes: Kidney, Tumor and Cyst. Compared with KiTS19 Challenge, cyst is a new semantic class, but these two tasks are quite close and that is why we choose nnUNet as our model and made some adjustments on it. Some important changes are made to the original nnUNet to adapt to this new task. Furthermore, we train models in 3 different ways and finally and merge them into one model by specific strategies. Detailed information is available in the part of Methods. The organizer uses an evaluation method called "Hierarchical Evaluation Classes" (HECs). The HEC scores of each model are showed in the following.

**Keywords:** semantic segmentation, nnU-Net, model ensemble

## 1 Introduction

This challenge is a semantic segmentation task of renal tumors and surrounding anatomy. The organizer provides 300 cases who undergone a contrast-enhanced preoperative CT scan that includes the entirety of all kidneys. Each case's most recent corticomedullary preoperative scan was (or will be) independently segmented three times for each instance of the following semantic classes—Kidney, Tumor, Cyst. Each instance was annotated by three independent people and final label is result of aggregating all of these files by various methods--OR, AND, MAJ. Another knowledge we use is that the scalar value of cysts in CT image is at the low level, since the cysts contain mostly water. This makes the scalar data augmentation not perform well.

Following contents in this paper:

**Methods**: In this part, we introduce our main approach and detailed parameters of our model.

**Results**: Official evaluation criteria is explained at first and then our results and training details is showed.

**Discussion and Conclusion**: we summarize our approach and results here, and point out the parts that can be improved.

## 2 Methods

We take nnU-Net as our main method and make some adjustments on it. Firstly, we adjust the data augmentation to adapt the challenge. We removed the scalar value related data augmentation strategy, as it modifies the scalar value. Secondly, we use a "two-step" method to accomplish the whole task. We put tumors and cysts into one class to train a model in the first part, and trained another model to distinguish cyst from tumor in the second part. Thirdly, we trained another model which takes kidney and cyst as a whole. At last, we take advantage of model ensemble to integrate trained models into a system. We use the "two-step" method and the traditional "one-step" three-classification method respectively, and integrate the 3 methods as our final model.

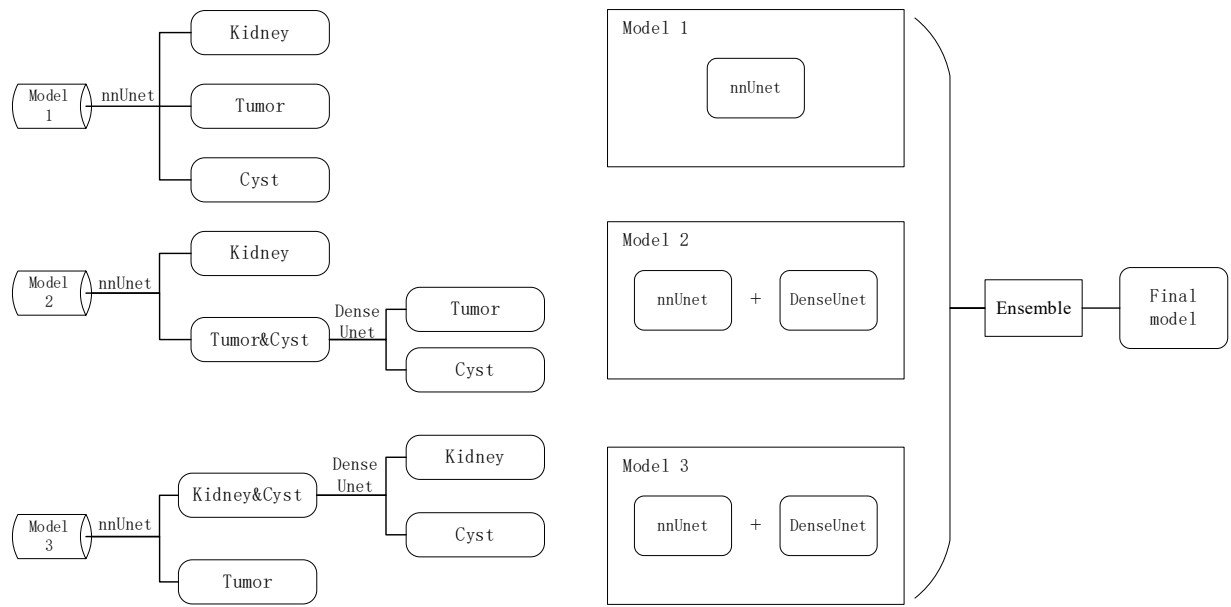

*Fig. 1: The brief introduction of our main approach. Three parallel models are trained and ensembled into a final model. In fact, this simple way has a good effect on the result. DenseUNet is used here for that the lightweight net takes short time to train but has a similar effect at the same time.*

## 2.1 Training and Validation Data

Our submission made use of the official KiTS21 training set alone.

## 2.2 Preprocessing

As the first processing step, nnU-Net crops the provided training cases to their nonzero region. It reduced the image size of datasets substantially and thus improved computational efficiency.

**Resampling** In the medical domain, the voxel spacing (the physical space the voxels represent) is heterogeneous. To cope with this heterogeneity, nnU-Net resamples all images to the same target spacing using either third order spline, linear or nearest neighbor interpolation. In our project, the original image size is $512 \times 512$ and finall resampled to $603 \times 603$.

**Data augmentation** A variety of data augmentation techniques are applied on the fly during training: rotations, scaling, Gaussian noise, Gaussian blur, brightness, simulation of low resolution, gamma and mirroring. Gray enhancement is discontinued because it changes the gray value which is important to the recognition of cysts.

**ROI** It is a part only for DenseUnet. Taking advantage of the preliminary segmentation results of nnUnet, we work out the ROI(region of interest) for DenseUnet. It proves that this action will help reduce scale of DenseUnet and improve the result.

## 2.3 Proposed Method

**Network architecture** All U-Net architectures configured by nnU-Net originate from the same template. This template closely follows the original U-Net[3] and its 3D counterpart. There are not many changes compared with traditional U-Net. Batch normalization, which is often used to speed up or stabilize the training, does not perform well with small batch sizes. Therefore, nnU-Net use instance normalization for all U-Net models. And the default kernel

size for convolutions is $3 \times 3 \times 3$ and $3 \times 3$ for 3D U-Net and 2D U-Net, respectively.

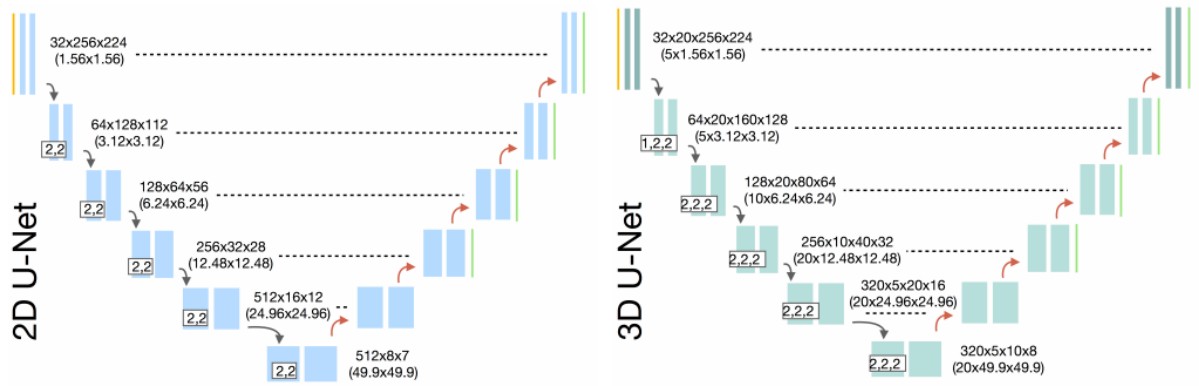

Fig. 2: The network structure of nnUnet including 3D U-Net and 2D U-Net.

The original author of DenseUnet[4] use it on automatic liver and tumor segmentation. DenseUnet applies the technique of densenet to U-net and has a good effect on the problem of segmentation. For each 3D input, the volume of 3D is quickly reduced to adjacent slices of 2D through the transform processing function F proposed in this paper. These 2D slices are then fed into the 2D DenseUNet to extract intra-chip features.

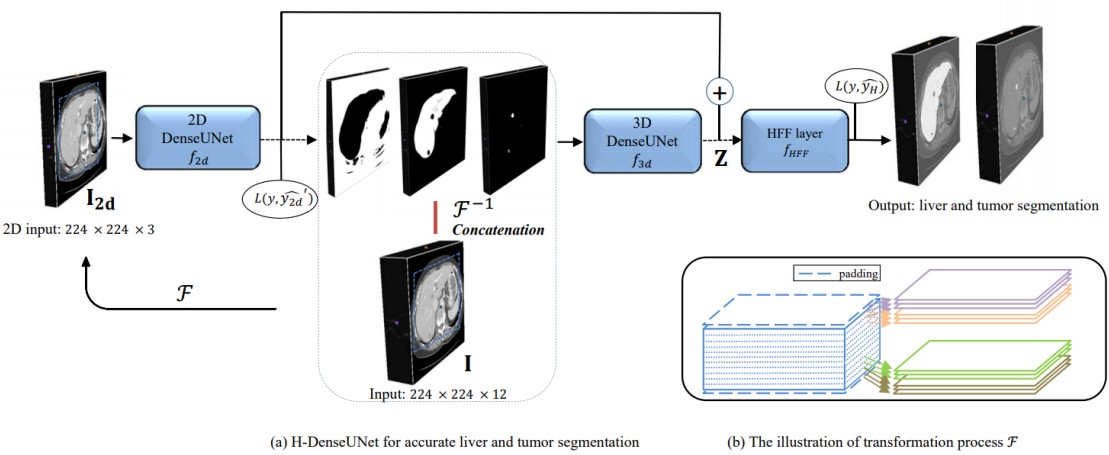

(a) H-DenseUNet for accurate liver and tumor segmentation     (b) The illustration of transformation process $\mathcal{F}$

Fig. 3: The network structure of DenseUnet. The original input image size is 224×224×3, in our project, the size is modified to 512×512×3. And the other parameters are the same.

**Loss function** The loss function is the sum of cross-entropy and Dice loss. For each deep supervision output, a corresponding down-sampled ground truth segmentation mask is used for loss computation.

**Optimization strategy** We simply use the Adam optimizer provided by PyTorch.

**Validation and ensembling strategy** We put each image into models trained by different methods. And to validate them, we determine the final prediction by majority voting.

**Post-processing** Connected component-based postprocessing[2] is commonly used in medical image segmentation.

Especially in organ segmentation it often helps to remove spurious false positive detections by removing all but the largest connected component. nnUNet follows this assumption and automatically benchmarks the effect of suppressing smaller components on the cross-validation results. It can be explained by the picture below.

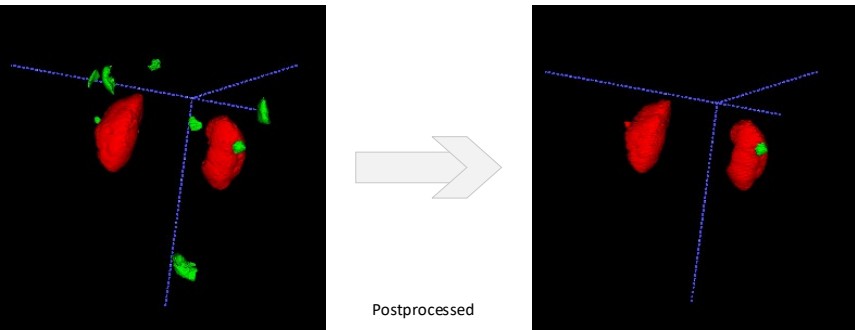

Postprocessed

*Fig. 4: The demonstration of the effect of Post-processing. By taking advantage of connected component-based postprocessing, we eliminate noises around the kidney area.*

## 3   Results

**Evaluative criteria** The organizer uses an evaluation method called "Hierarchical Evaluation Classes" (HECs). In an HEC, classes that are considered subsets of another class are combined with that class for the purposes of computing a metric for the superset. HECs: 1. Kidney and Masses:(Kidney + Tumor + Cyst) 2. Kidney Mass:(Tumor + Cyst) 3. Tumor:(Tumor only).

**Results** Here are the scores of our models and some living examples
(1) HEC scores
    a. model 1 (on validation set)

|  | Sørensen-Dice | Surface Dice |
| --- | --- | --- |
| Kidney and Masses | 0.94041 | 0.89721 |
| Kidney Mass | 0.83983 | 0.74429 |
| Tumor | 0.83207 | 0.72894 |

  b. model 2 (on validation set)

|  | Sørensen-Dice | Surface Dice |
| --- | --- | --- |
| Kidney and Masses | 0.93036 | 0.89275 |
| Kidney Mass | 0.83049 | 0.72003 |
| Tumor | 0.76701 | 0.66083 |

  c. model 3 (on validation set)

|  | Sørensen-Dice | Surface Dice |
| --- | --- | --- |
| Kidney and Masses | 0.92845 | 0.89135 |
| Kidney Mass | 0.77596 | 0.61060 |
| Tumor | 0.79925 | 0.68406 |

  d. final model (waiting for official results)

(2) Some examples of predictions next to human-labels

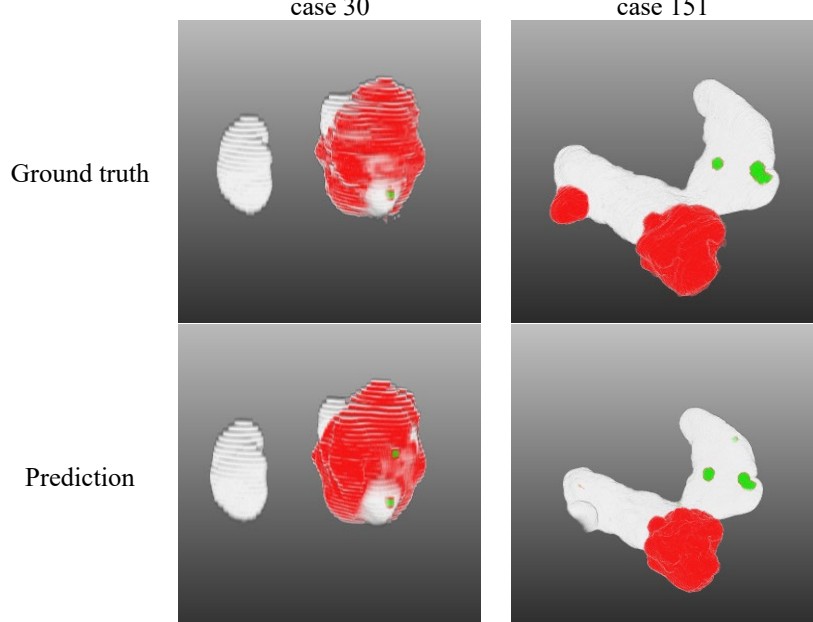

*Fig. 5: examples of case 30 and case 151.*

**Training details** Based on experience and as a trade-off between runtime and reward, all networks are trained for 1000 epochs with one epoch being defined as iteration over 250 minibatches. It took 15 days to train each model and the project took 2 months in total.

## 4  Discussion and Conclusion

Our project mainly takes advantage of current efficient network—nnU-Net and does not make a lot of innovation. But the power of nnU-Net is so strong and it is not easy to find a better way. There is a lot of room for improvement in our result because all our members are newbies in this field, and the time for this task is urgent. In the future, we can also try some other networks and compare the effect with the previous method.

All in all, thank you very much to the organizer for providing this opportunity. Although our result is not so perfect, we have harvested a lot in the participation.

## Acknowledgements

Thank MIC-DKFZ[5] very much for nnUnet.
Thank nitsaick[6] very much for DenseUnet.

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
