# OpenReview forum: "Modified nnU-Net for the MICCAI KiTS21 Challenge"
_MICCAI.org/2021/Challenge/KiTS — Submitted to KiTS21 Challenge_

### Official Review · Reviewer_4BwE · 2021-08-30

**Rating:** 6

**Review:**

The authors present an "out-of-the-box nnU-Net" approach to this problem. They do an adequate job describing what that entailed but their paper could benefit from a greater level of detail. Currently, they offload too much of the methods description to the nnU-Net reference, whereas the reader should be able to understand generally what was done by reading this paper alone. Therefore, the authors should aim to expand their methods section. Adding more text to the introduction and supporting more of their statements with references would also be helpful -- especially references to the published literature and not just to a URL.

---

### Official Review · Reviewer_4gdN · 2021-08-30

**Rating:** 5

**Review:**

### Overall

- I'm not sure that it is allowed to have different paragraphs within the abstract, and the abstract is a little bit long. It would be good to concatenate into a single paragraph and move some of the details to elsewhere in the paper.

### Introduction

- It might be nice to include a short section at the end of the introduction that summarizes each section for the rest of the paper

### Methods

- What dimensions did you resample your images to?
- Did you clip your intensity values? If so, to what? How did you approach normalization? I understand you used the default behavior of nnU-Net, but what value did it use?
- Pleas expand on what you mean by your statement "nnUNet follows this assumption and automatically benchmarks the effect of suppressing smaller components on the cross-validation results"

### Results

- Please include your official results once they are known

### Discussion and Conclusion

- This section should be numbered 4 instead of 3 (this may have been an error with the template)

---

### Decision · Program_Chairs · 2021-08-30

**Decision:**

Major Revisions

**Comment:**

Please address the reviewer comments and resubmit